# Minimum Alveolar Concentration of Isoflurane in Rats Chronically Treated with the Synthetic Cannabinoid WIN 55,212-2

**DOI:** 10.3390/ani12070853

**Published:** 2022-03-29

**Authors:** Julio Raul Chavez-Monteagudo, José Antonio Ibancovichi, Pedro Sanchez-Aparicio, Sergio Recillas-Morales, Jorge Osorio-Avalos, Marco Antonio De Paz-Campos

**Affiliations:** 1Department of Veterinary Anesthesia and Analgesia, Facultad de Estudios Superiores Cuautitlán, Hospital de Pequeñas Especies, Universidad Nacional Autónoma de México, Cuautitlan Izcalli 54740, Mexico; juliochavezmonteagudo@comunidad.unam.mx; 2Department of Veterinary Anesthesia and Analgesia, Faculty of Veterinary Medicine, Universidad Autónoma del Estado de México, Toluca 50000, Mexico; 3Department of Pharmacology, Faculty of Veterinary Medicine, Universidad Autónoma del Estado de México, Toluca 50000, Mexico; pedrosanchezaparicio0@gmail.com (P.S.-A.); srecillasm@uaemex.mx (S.R.-M.); 4Department of Biostatistics, Faculty of Veterinary Medicine, Universidad Autónoma del Estado de México, Toluca 50000, Mexico; josorioa@uaemex.mx; 5Department of Pharmacology, Facultad de Estudios Superiores Cuautitlán, Hospital de Pequeñas Especies, Universidad Nacional Autónoma de México, Cuautitlan Izcalli 54740, Mexico; depaz_0@comunidad.unam.mx

**Keywords:** minimum alveolar concentration MAC, isoflurane, WIN 55,212-2, rats

## Abstract

**Simple Summary:**

The minimum alveolar concentration of isoflurane (inhaled anesthetic required to prevent movement in 50% of subjects exposed to a supramaximal noxious stimulus) was determined in 24 male rats chronically treated with the synthetic cannabinoid WIN 55,212-2 to evaluate the interaction of isoflurane with chronically administered cannabinoid agonist. The minimum alveolar concentration was determined in one group without treatment, in rats treated for 21 days with WIN 55,212-2, and another group 8 days after stopping treatment for 21 days with cannabinoid. We believe it is necessary to study the effects of chronic consumption of these substances on the requirements of inhalation anesthetics in patients that will be submitted to general anesthesia. The administration for 21 days of WIN 55,212-2 increases the minimum alveolar concentration of isoflurane in rats; this effect does not disappear after 8 days of discontinuing treatment with the synthetic cannabinoid.

**Abstract:**

The minimum alveolar concentration MAC of isoflurane was measured in rats chronically treated with WIN 55,212-2. Methods: The MAC of isoflurane was determined in 24 male rats from expiratory samples at time of tail clamping under the following conditions: without treatment MAC(_ISO_), in rats treated for 21 days with WIN 55,212-2 MAC(_ISO + WIN55_), and in rats 8 days after stopping treatment with WIN 55,212-2 (MAC_ISO + WIN55 + 8D_). Results: The MAC(_ISO_) was 1.32 ± 0.06. In the MAC(_ISO + WIN55_) group, the MAC increased to 1.69 ± 0.09 (28%, *p*-value ≤ (0.0001). Eight days after stopping treatment with WIN55, the MAC did not decrease significantly, 1.67 ± 0.07 (26%, *p*-value ≤ 0.0001). Conclusions: The administration of WIN 55,212-2 for 21 days increases the MAC of isoflurane in rats. This effect does not disappear 8 days after discontinuation of treatment with the synthetic cannabinoid.

## 1. Introduction

*Cannabis sativa*, one of the oldest psychotropic drugs known [1], is also the most consumed drug according to the World Drug Report 2019, published by the United Nations, which estimated that there are over 188 million consumers of cannabis [2]. 

Delta-9-tetrahydrocannabinol (Delta 9-THC) was isolated in the 1960s [3], the cannabinoid type 1 receptor (CB1) was identified at the end of the 1980s [4], and the cannabinoid type 2 receptor (CB2) was discovered in 1993 [5]. Subsequently, the first [6] and second endogenous cannabinoids were discovered [7]. The endocannabinoid system (ECS) is involved in many health and disease processes [8], and different research groups have performed investigations [9] by modulating the activity of the ECS and assessing the possible effects of manipulating this system. This information could have important therapeutic applications [10,11,12] in a variety of diseases, such as epilepsy and acute and chronic pain in humans.

Therefore, we believe it is necessary to study the effects of chronic consumption of these substances on the requirements of inhalation anesthetics in patients that will be submitted to general anesthesia. 

The objective of this study was to evaluate the interaction of a chronically administered cannabinoid agonist on the minimum alveolar concentration (MAC_ISO_); to the knowledge of the authors, there are no studies that evaluate this interaction. MAC is defined as the minimum alveolar concentration of an inhaled anesthetic required to prevent movement in 50% of subjects exposed to a supramaximal noxious stimulus and represents an index of the potency of anesthetic agents [13]. The authors hypothesize that cannabinoids administered chronically have a different effect than that previously reported in acute administration of MAC.

## 2. Materials and Methods

This experiment was approved by the Animal Research Ethics Committee for Animal Experimentation of the Faculty of Veterinary Medicine of the University of the State of Mexico (protocol number 3492/2013CHT). A total of 24 male Wistar rats weighing 310 ± 20 g were used. 

The rats were housed in groups of 4, in Plexiglas cages, with a 12 h light/12 h dark cycle (lights on at 07:00), with a relative humidity of 50–60% and ambient temperature of 23 ± 2 °C. The animals had free access to water and rodent food (Prolab1 RMH 2500, St. Louis, MO, USA). Animals were allowed to acclimatize for one week before the experiments took place. These experiments were performed during the morning (09:00–12:00). All animals were handled according to the guidelines in the Guide for the Care and Use of Laboratory Animals [14].

### 2.1. Anesthetic Procedure

Anesthesia was induced by placing each rat in the induction chamber and delivering 5% isoflurane (Forane; Baxter Laboratories, Irvine, CA, USA) with a continuous oxygen flow rate of 5 L/min. Once the animal was adequately anesthetized, it was removed from the induction chamber and placed in dorsal recumbency for endotracheal intubation. The oral cavity was opened, and, with the aid of a laryngoscope, the larynx was visualized, and a flexible blunt-tip wire guide was inserted and used to direct the endotracheal catheter (16G Teflon catheter: Introcan; B-Braun, Sao Goncalo, Brazil), which was secured to the maxilla by means of adhesive tape. 

Correct placement of the catheter was confirmed by CO_2_ infrared absorption analysis (BeneView T5, Mindray, Multi-gas offers, Shenzhen, China). The catheter was connected to a small T-piece breathing system with minimal dead space and fresh gas flow of 1 L/min of oxygen. The isoflurane concentration was adjusted as necessary based on an assessment of the palpebral reflex and hemodynamic responses during instrumentation. During the study, rats were breathing spontaneously. 

The carotid artery was exposed surgically and catheterized using a 24- gauge catheter (Introcan; B-Braun, São Gonçalo, Brazil). This catheter was connected to a pressure transducer system for direct blood pressure monitoring and the collection of arterial blood to determine blood gases. Systolic, diastolic, and mean arterial blood pressures (SAP, DAP, and MAP, respectively) and heart rate (HR) were continuously monitored (BeneView T5, Mindray, Shenzhen, China). For blood gas analysis (GEM Premier 3000; Instrumentation Laboratory, Seattle, WA, USA), 0.3 mL of blood was obtained immediately after determining the MAC to ensure (at that time) that values were within normal physiological parameters. Rectal temperature was maintained between 37 °C and 38 °C by means of a convective warming system (Equator1, SurgiVet1, Smiths Medical PM Inc., San Clemente, CA, USA). Inspired isoflurane (Fi_Iso_), end-tidal (Fe_Iso_) concentrations, end-tidal carbon dioxide tension (PEtCO_2_), and respiratory rate (RR) were continuously measured with an infrared gas analyzer previously calibrated (BeneView T5, Mindray, multi-gas offers, Shenzhen, China) by endotracheal gas sampling (60 mL/min) obtained by means of a catheter inserted through the endotracheal tube with the tip located at the level of the carina. 

### 2.2. MAC Determination

Once instrumentation was performed, and prior to assessing MAC isoflurane, Fe_Iso_ was adjusted to 1.32%, which is a value close to the isoflurane MAC previously reported by the authors [15]. Once this concentration was achieved, it was maintained for 15 min in order to achieve the equilibrium of isoflurane partial pressure between alveolar gas, arterial blood, and the spinal cord. [16] The isoflurane MAC was determined by the tail clamp method described by Quasha et al. [17]. A painful noxious stimulus was applied with a hemostat clamped (8-inch Rochester Dean hemostatic forceps) on the tail at a specific end-tidal concentration of each volatile agent. The tail was clamped to the first ratchet lock for 60 s or until a positive response was observed. The tail was always stimulated proximally to the previous test site. A positive motor response was considered if jerking or twisting motions of the head or body, or movement of the extremities was observed. Negative responses included a lack of movement of the head and limbs, muscle rigidity, shivering, swallowing, and chewing; movement of the tail should not be considered.

If the response was positive, the delivered volatile anesthetic concentration was increased by 10%, and, if the response was negative, the concentration of the volatile anesthetic was decreased by 10%. After an equilibration period of 15 min, the application of the stimulus was repeated. The person assessing the response was blinded with respect to the drugs administered to each rat. In each rat, the MAC was evaluated in duplicate. 

The MAC of isoflurane values was corrected to sea level by use of the formula (barometric pressure of location/760 mmHg) x obtained MAC value. The mean barometric pressure was obtained from the official city meteorological station for the altitude at which the experiment was performed (2680 m above sea level) and was 556 mmHg. At the end of each experiment, animals were euthanized with pentobarbital given intravenously (Anestesal, Pfizer, Toluca, Mexico) to animals deeply anaesthetized with the inhalant agent. 

### 2.3. Experimental Design

Using a random number generator (Excel 2007, Microsoft Office), the animals were distributed into three groups (*n* = 8). 

The control group MAC(_ISO_) remained untreated for the measurement.

The MAC (_ISO + WIN55_) group was treated intraperitoneally (i.p) with 1 mg/kg of WIN 55,212-2 (mesylate salt, Sigma-Aldrich, St. Louis, MO, USA) every 24 h (at 09:00 h) for 21 days, Lawston et al. [18]. 

WIN 55,212-2 was suspended in a vehicle solution of 0.3% Tween 80 in saline (0.9%), as described by Tanda et al. [19]. Isoflurane MAC measurements were performed 24 h after the last treatment of WIN 55,212-2 (day 22). 

The MAC (_ISO + WIN55 + 8D_) group was treated i.p with 1 mg/kg of WIN 55,212-2 every 24 h (at 09:00 h) for 21 days. The measurement of isoflurane MAC was performed 8 days after the last treatment with WIN 55,212-2 (day 29).

### 2.4. Statistical Analysis

Statistical analysis was performed using Prism 6 (GraphPad Software, Inc., San Diego, CA, USA). The Shapiro–Wilk test was used for the assessment of data normality. Data are reported as mean ± standard deviation (SD). Analysis of variance was performed, and post hoc comparison of the groups was performed using the Holm–Sidak test. Values were considered statistically different when *p* < 0.05. 

## 3. Results

The results are summarized in Table 1. The value of the mean ± SD obtained in the MAC_ISO_ group was 1.32% ± 0.06. The MAC_ISO + WIN55_ group showed a 28% increase in the MAC isoflurane; the mean value for this group was 1.69% ± 0.09 and was significantly different when compared to the MAC_ISO_ group (*p* < 0.0001). The MAC value of the MACISO + WIN55 + 8D group was 1.67% ± 0.07, and there was no difference when compared to the MACISO + WIN55 group (*p* = 0.6995), but a significant difference was found when compared with the MACISO (*p* < 0.0001). Table 2 shows the cardiorespiratory and temperature values of the different study groups in which no significant statistical differences are observed between the different values.

## 4. Discussion

In this work, we observed that a synthetic cannabinoid chronically administered increases the MAC of isoflurane. 

While the use of cannabinoids has increased in a recreational and therapeutic way [20], to the knowledge of the authors, there are no reports of the effect of chronic administration of cannabinoids on the requirements of inhalation anesthetics. In this investigation, we observed that after 21 days of administering the synthetic cannabinoid WIN 55,212-2, MAC of isoflurane in rats increased. However, this observation may not necessarily reflect the effect that occasional *Cannabis sativa* consumption could generate in humans. The most abundant substance present in the cannabis plant, Δ9-THC, is responsible for its psychotropic effects [2] and is a phytocannabinoid and partial agonist of the CB1 receptor. WIN 55,212-2 is a synthetic total cannabinoid agonist of the CB1/CB2 receptors [20]; therefore, the effectiveness of the cannabinoid agonist and the duration of exposure could influence the effect on the requirements of inhalation anesthetics.

WIN 55,212-2 causes an increase in the central levels of norepinephrine. Page and collaborators [21] showed that rats treated with WIN 55,212-2 for 8 days had an increase in noradrenergic activity. Furthermore, they demonstrated that repeated administration of WIN 55,212-2 stimulates CB1 cannabinoid receptors in cell bodies of the locus coeruleus and nerve terminals containing norepinephrine, generating an increase in norepinephrine efflux.

Norepinephrine levels in nervous terminals modulate the MAC response of isoflurane as Miller et al. [22] demonstrated in a previous work, in experiments following the administration of alpha-methyldopa, reserpine, and iproniazid. These authors demonstrated that the requirements of inhalational anesthetics are related to norepinephrine levels. Thus, drugs that decrease the concentration of norepinephrine in the central nervous system decrease the MAC. On the other hand, drugs that increase norepinephrine levels cause an increase in the requirement for inhalation anesthetics. Similarly, it has been reported that acute administration of amphetamine [23] and cocaine [24] increases the MAC of halothane in dogs due to an increase in the catecholamine concentration in the central nervous system.

In addition, diverse neurotransmitters, such as norepinephrine, may manifest some of their actions by strongly inhibiting TWIK-related acid-sensitive K+ channels (TASK) and thus influence neuronal excitability. Similarly, inhalation anesthetics activate, and cannabinoid agonists inhibit, TASK channels [25,26]. Perhaps the collective inhibition between norepinephrine and cannabinoids of TASK channels led to the need for a higher concentration of isoflurane to increase the anesthetic requirements necessary to prevent movement in response to painful stimulus. 

Previous studies have reported that acute and subacute administration of Δ9-THC decreases the MAC of halothane in dogs [27], and the MAC cyclopropane decreases in rats [28] when treated acutely and chronically (for one week) with Δ9-THC. This discrepancy with our results may be explained on the basis of the experimental design, while we administered the cannabinoid receptor agonist for 21 days, Stoelting and Vitez administered the treatment for a shorter period of time, besides we used WIN55,212-2, while the cited works used Δ9-THC. 

Another explanation for the differences observed in MAC isoflurane can be through the observations made by Mechoulam et al. [29] and by Marciano et al. [30], which indicate that endocannabinoids have paradoxical effects on the central nervous system of mammals since, in some cases, they generate an increase in neuronal excitability and, in others, they decrease it, depending on the dose administered. Similarly, they reported that cannabinoids cause short-term inhibitory effects on the release of glutamate; in contrast, prolonged stimulation of CB1 receptors by exogenous administration of cannabinoids could block the release of the inhibitory neurotransmitter GABA [30]. 

In this sense, an increase in the MAC of isoflurane suggests that systemic and sustained administration of WIN 55,212-2 reflected an increase in the anesthetic requirements necessary to prevent movement in response to painful stimulus. 

When measuring MAC after 8 days of stopping cannabinoid administration (MAC_ISO + WIN55 + 8D_ group), we found no statistically significant differences compared to the MAC_ISO + WIN55_ group. Therefore, the increase in the isoflurane MAC caused by the administration of WIN 55,212-2 does not decrease after 8 days of stopping the administration of the cannabinoid, possibly caused by an increase in noradrenergic activity sustained. This difference with the studies carried out by Page and collaborators [21], where a decrease in activity was observed at 8 days, could be a consequence of the longer exposure to the cannabinoid used in our study and possibly it will take more time for the decrease in noradrenergic activity since we administered it for 21 days. [31]. To determine whether the effect of WIN 55,212-2 on isoflurane MAC is transient, further experiments will be necessary where different administration intervals, as well as different doses, are evaluated since this could be considered a limitation of our study where we use a single dose.

## 5. Conclusions

The administration for 21 days of WIN 55,212-2 increases the MAC of isoflurane in rats; this effect does not disappear after 8 days of discontinuing treatment with the synthetic cannabinoid.

## Figures and Tables

**Table 1 animals-12-00853-t001:** MAC of Isoflurane in rats chronically treated with the synthetic cannabinoid WIN 55,212-2.

Group	MAC%	SD	% MAC Increase	*p*-Value	95% IC
MAC_ISO_	1.32	0.06		-	1.27–1.37
MAC_ISO + WIN55_	1.69 *	0.09	28%	<0.0001	1.58–1.77
MAC_ISO + WIN + 8D_	1.67 *	0.07	26%	<0.0001	1.60–1.75

* Statistically significant compared to the control group CAM_ISO_ (*p* < 0.05).

**Table 2 animals-12-00853-t002:** Cardiorespiratory and temperature values of the different study groups.

Value	MAC_ISO_	MAC_ISO + WIN55_	MAC_ISO + WIN + 8D_
Hearth rate (bpm)	401 ± 8	403 ± 7	403 ± 9
Mean arterial blood pressure (mmHg)	93 ± 8	90 ± 9	91 ± 6
Temperature °C	37.7 ± 0.07	37.6 ± 0.12	37.7 ± 0.06
pH	7.3 ± 03	7.3 ± 0.04	7.3 ± 0.03
PaO_2_ (mmH)	301 ± 34	295 ± 8	288 ± 28
PaCo_2_ (mmHg)	37 ± 4	37 ± 1	37 ± 1

## Data Availability

The data presented in this study are available on request from the corresponding author.

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
