# Peer review of "Minimum Alveolar Concentration of Isoflurane in Rats Chronically Treated with the Synthetic Cannabinoid WIN 55,212-2"

_animals, 2022, doi:10.3390/ani12070853_

Round 1
Reviewer 1 Report
Can you also include a comment on the dose of cannabinoid used in your study or a comment that a limitation of the study is that a single dose was used.
Minimum alveolar concentration of isoflurane in rats chronically treated with the synthetic cannabinoid WIN 55,212-2
Suggestions:
Add a full stop at the end of the title
Change from ‘isoflurane MAC’ to ‘MAC of isoflurane throughout’
Change from ‘…the end expiratory samples at time of tail clamping under…’ to ‘…end expiratory samples at the time of tail clamping under…’
Change ‘…and other group 8 days after stopping treatment for 21 days with WIN 5…’ to ‘…and in rats 8 days after stopping treatment with WIN 5…’
Change ‘Results: The MACISO was 1.32 ± 0.06. In the MACISO+WIN55 group the MAC increase to 1.69 ± 0.09 (28%). After 8 days stopping treatment, MAC did not decrease significantly 1.67 ± 0.07 (26%). Conclusions: The administration for 21 days of WIN 55,212-2 increases the MAC of isoflurane in rats; this effect does not disappear after 8 days of discontinuing treatment with the synthetic cannabinoid.’ To Results: The MACISO was 1.32 ± 0.06. In the MACISO+WIN55 group the MAC increased to 1.69 ± 0.09 (28%, p value = xxx). Eight days after stopping treatment with WIN55, the MAC did not decrease significantly 1.67 ± 0.07 (26%, p value = xxx). Conclusions: The administration of WIN 55,212-2 for 21 days increases the MAC of isoflurane in rats. This effect does not disappear 8 days after discontinuation of treatment with the synthetic cannabinoid.
Change ‘was discovered in 1993 (5); subsequently, the first (6) and second endogenous cannabinoids were discovered (7).’ To ‘was discovered in 1993 (5). Subsequently, the first (6) and second endogenous cannabinoids were discovered (7).’
Change ‘The objective of this study was to evaluate the interaction of chronically administered cannabinoid agonist on the minimum alveolar concentration…’ to ‘The objective of this study was to evaluate the interaction of a chronically administered cannabinoid agonist on the minimum alveolar concentration …’
Change ‘In this study, with protocol number 3492/2013CHT, 24 male Wistar rats weighing 310 ± 20 g was used. This experiment was approved by the Animal Research Ethics Committee for Animal Experimentation of the Faculty of Veterinary Medicine of the University of the State of Mexico.’ To ‘This experiment was approved by the Animal Research Ethics Committee for Animal Experimentation of the Faculty of Veterinary Medicine of the University of the State of Mexico (protocol number 3492/2013CHT). Twenty four male Wistar rats weighing 310 ± 20 g were used.’
Provide reference for the Guide for the care and Use of laboratory Animals.
Change’ Anesthetic induction was performed by placing each rat in the induction chamber providing 5% isoflurane (Forane; Baxter Laboratories, USA) with a continuous oxygen flow of 5 L/min. Once the animal was anesthetized and lost jaw tone, it was removed from the induction chamber and placed in dorsal recumbence for endotracheal intubation. The oral cavity was opened and with the aid of an otoscope, the larynx was visualized and a flexible blunt-tip wire guide was inserted inside and used to direct the endotracheal catheter (16G Teflon catheter: Introcan; B-Braun, Brazil), which was secured to the maxilla by means of adhesive tape.’ To ‘Anesthesia was induced by placing each rat in the induction chamber and delivering 5% isoflurane (Forane; Baxter Laboratories, USA) with a continuous oxygen flow of 5 L/min. Once the animal was adequately anesthetized, it was removed from the induction chamber and placed in dorsal recumbency for endotracheal intubation. The oral cavity was opened and with the aid of an otoscope, the larynx was visualized and a flexible blunt-tip wire guide was inserted and used to direct the endotracheal catheter (16G Teflon catheter: Introcan; B-Braun, Brazil), which was secured to the maxilla by means of adhesive tape.’
Change ‘The carotid artery was exposed via surgical cut-down and catheterized using a 24- gauge catheter (Introcan; B-Braun, Brazil); this was connected to a pressure transducer system for direct blood pressure monitoring and the collection of …’ to ‘The carotid artery was exposed surgically and catheterized using a 24- gauge catheter (Introcan; B-Braun, Brazil). This catheter was connected to a pressure transducer system for direct blood pressure monitoring and the collection of …’
Change ‘The mean value of the MACISO+WIN55+8D group was 1.67% ± 0.07; there was no statistically difference when compared to the MACISO+WIN55 group (p=0.6995), but a significant difference was found compared with the MACISO (p<0.0001).’ to ‘The MAC value of the MACISO+WIN55+8D group was 1.67% ± 0.07 and there was no difference when compared to the MACISO+WIN55 group (p=0.6995), but a significant difference was found when compared with the MACISO (p<0.0001).’
Change ‘MAC is defined as the alveolar concentration of an inhalatory anesthetic necessary to prevent movement in 50% of subjects exposed to a supramaximal noxious stimulus and represents an index of potency of anesthetic agents’ to ‘MAC is defined as the minimum alveolar concentration of an inhaled anesthetic required to prevent movement in 50% of subjects exposed to a supramaximal noxious stimulus and represents an index of potency of anesthetic agents’
Change ‘The use of cannabinoids has increased in a recreational and therapeutic way (2), to the knowledge of the authors, there are no reports of the effect of chronic administration of cannabinoids on the requirements of inhalation anesthetics.’ To ‘ While the use of cannabinoids has increased in a recreational and therapeutic way (2), to the knowledge of the authors, there are no reports of the effect of chronic administration of cannabinoids on the requirements of inhalation anesthetics.’
Change ‘this investigation, we observed that 21 days of administering the synthetic cannabinoid WIN 55,212-2 increased the isoflurane MAC in rats’ to ‘this investigation, we observed that after 21 days of administering the synthetic cannabinoid WIN 55,212-2 MAC of isoflurane in rats increased.’
Change ‘WIN 55,212-2 cause an increase in the central levels of norepinephrine, Page and collaborators (20) showed that rats treated with WIN 55,212-2 for 8 days presented an increase in noradrenergic activity, furthermore they demonstrated that repeated administration of WIN 55,212-2 stimulates CB1 cannabinoid receptors in cell bodies of the locus coeruleus and in nerve terminals containing norepinephrine, generating an increase in norepinephrine efflux.’ To ‘WIN 55,212-2 causes an increase in the central levels of norepinephrine: Page and collaborators (20) showed that rats treated with WIN 55,212-2 for 8 days had an increase in noradrenergic activity. Furthermore they demonstrated that repeated administration of WIN 55,212-2 stimulates CB1 cannabinoid receptors in cell bodies of the locus coeruleus and in nerve terminals containing norepinephrine, generating an increase in norepinephrine efflux.’
Change ‘Norepinephrine levels in nervous terminals modulates MAC response of isoflurane as Miller et al’ to ‘Norepinephrine levels in nervous terminals modulate the MAC response of isoflurane as Miller et al’
Change ‘demonstrated in a previous work, in experiments following the administration of alpha-methyldopa, reserpine and iproniazid, they demonstrated that the requirements of inhalational anesthetics are related to norepinephrine levels.’ To ‘demonstrated in a previous work, in experiments following the administration of alpha-methyldopa, reserpine and iproniazid. These authors demonstrated that the requirements of inhalational anesthetics are related to norepinephrine levels.’
Change ‘It´s also known, diverse neurotransmitters such as norepinephrine, might manifest some of their actions by strongly inhibiting TWIK-related acid-sensitive K+ channels (TASK) and thus influence neuronal excitability.’ To ‘In addition, diverse neurotransmitters such as norepinephrine, may manifest some of their actions by strongly inhibiting TWIK-related acid-sensitive K+ channels (TASK) and thus influence neuronal excitability.’
Change ‘Similarly, Inhalation anesthetics activate and cannabinoid agonists inhibit TASK channels’ to ‘Similarly, inhalation anesthetics activate and cannabinoid agonists inhibit, TASK channels’
Change halothane MAC and cyclopropane MAC to MAC of halothane and MAC of cyclopropane
Change ‘for a lees period of time’ to ‘for a shorter period of time.’
Change ‘observed in de isoflurane MAC’ to ‘observed in MAC of isoflurane’
Author Response
Can you also include a comment on the dose of cannabinoid used in your study or a comment that a limitation of the study is that a single dose was used.
Thank you for your comment.
We have placed the following comment in the last paragraph of the discussion.
To determine whether the effect of WIN 55,212-2 on isoflurane MAC is transient, further experiments will be necessary where different administration intervals as well as different doses are evaluated, since this could be considered a limitation of our study where we use a single dose.
Minimum alveolar concentration of isoflurane in rats chronically treated with the synthetic cannabinoid WIN 55,212-2
Suggestions:
Add a full stop at the end of the title
Thanks for your comment it has been corrected
Change from ‘isoflurane MAC’ to ‘MAC of isoflurane throughout’
Thanks for your comment it has been corrected
Change from ‘…the end expiratory samples at time of tail clamping under…’ to ‘…end expiratory samples at the time of tail clamping under…’
Thank you for your comment. The paragraph has been changed based on your suggestion
Change ‘…and other group 8 days after stopping treatment for 21 days with WIN 5…’ to ‘…and in rats 8 days after stopping treatment with WIN 5…’
Thank you for your comment. The paragraph has been changed based on your suggestion
Change ‘Results: The MACISO was 1.32 ± 0.06. In the MACISO+WIN55 group the MAC increase to 1.69 ± 0.09 (28%). After 8 days stopping treatment, MAC did not decrease significantly 1.67 ± 0.07 (26%). Conclusions: The administration for 21 days of WIN 55,212-2 increases the MAC of isoflurane in rats; this effect does not disappear after 8 days of discontinuing treatment with the synthetic cannabinoid.’ To Results: The MACISO was 1.32 ± 0.06. In the MACISO+WIN55 group the MAC increased to 1.69 ± 0.09 (28%, p value = xxx). Eight days after stopping treatment with WIN55, the MAC did not decrease significantly 1.67 ± 0.07 (26%, p value = xxx). Conclusions: The administration of WIN 55,212-2 for 21 days increases the MAC of isoflurane in rats. This effect does not disappear 8 days after discontinuation of treatment with the synthetic cannabinoid.
Thank you for your comment. The paragraph has been changed based on your suggestion
Change ‘was discovered in 1993 (5); subsequently, the first (6) and second endogenous cannabinoids were discovered (7).’ To ‘was discovered in 1993 (5). Subsequently, the first (6) and second endogenous cannabinoids were discovered (7).’
Thank you for your comment. The paragraph has been changed based on your suggestion
Change ‘The objective of this study was to evaluate the interaction of chronically administered cannabinoid agonist on the minimum alveolar concentration…’ to ‘The objective of this study was to evaluate the interaction of a chronically administered cannabinoid agonist on the minimum alveolar concentration …’
Thank you for your comment. The paragraph has been changed based on your suggestion
Change ‘In this study, with protocol number 3492/2013CHT, 24 male Wistar rats weighing 310 ± 20 g was used. This experiment was approved by the Animal Research Ethics Committee for Animal Experimentation of the Faculty of Veterinary Medicine of the University of the State of Mexico.’ To ‘This experiment was approved by the Animal Research Ethics Committee for Animal Experimentation of the Faculty of Veterinary Medicine of the University of the State of Mexico (protocol number 3492/2013CHT). Twenty-four male Wistar rats weighing 310 ± 20 g were used.’
Thank you for your comment. The paragraph has been changed based on your suggestion
Provide reference for the Guide for the care and Use of laboratory Animals.
Thanks for your comment the reference has been added
Change’ Anesthetic induction was performed by placing each rat in the induction chamber providing 5% isoflurane (Forane; Baxter Laboratories, USA) with a continuous oxygen flow of 5 L/min. Once the animal was anesthetized and lost jaw tone, it was removed from the induction chamber and placed in dorsal recumbence for endotracheal intubation. The oral cavity was opened and with the aid of an otoscope, the larynx was visualized, and a flexible blunt-tip wire guide was inserted inside and used to direct the endotracheal catheter (16G Teflon catheter: Introcan; B-Braun, Brazil), which was secured to the maxilla by means of adhesive tape.’ To Anesthesia was induced by placing each rat in the induction chamber and delivering 5% isoflurane (Forane; Baxter Laboratories, USA) with a continuous oxygen flow of 5 L/min. Once the animal was adequately anesthetized, it was removed from the induction chamber and placed in dorsal recumbency for endotracheal intubation. The oral cavity was opened and with the aid of an otoscope, the larynx was visualized, and a flexible blunt-tip wire guide was inserted and used to direct the endotracheal catheter (16G Teflon catheter: Introcan; B-Braun, Brazil), which was secured to the maxilla by means of adhesive tape.’
Thank you for your comment. The paragraph has been changed based on your suggestion
Change ‘The carotid artery was exposed via surgical cut-down and catheterized using a 24- gauge catheter (Introcan; B-Braun, Brazil); this was connected to a pressure transducer system for direct blood pressure monitoring and the collection of …’ to ‘The carotid artery was exposed surgically and catheterized using a 24- gauge catheter (Introcan; B-Braun, Brazil). This catheter was connected to a pressure transducer system for direct blood pressure monitoring and the collection of …’
Thank you for your comment. The paragraph has been changed based on your suggestion
Change ‘The mean value of the MACISO+WIN55+8D group was 1.67% ± 0.07; there was no statistically difference when compared to the MACISO+WIN55 group (p=0.6995), but a significant difference was found compared with the MACISO (p<0.0001).’ to ‘The MAC value of the MACISO+WIN55+8D group was 1.67% ± 0.07 and there was no difference when compared to the MACISO+WIN55 group (p=0.6995), but a significant difference was found when compared with the MACISO (p<0.0001).’
Thank you for your comment. The paragraph has been changed based on your suggestion
Change ‘MAC is defined as the alveolar concentration of an inhalatory anesthetic necessary to prevent movement in 50% of subjects exposed to a supramaximal noxious stimulus and represents an index of potency of anesthetic agents’ to ‘MAC is defined as the minimum alveolar concentration of an inhaled anesthetic required to prevent movement in 50% of subjects exposed to a supramaximal noxious stimulus and represents an index of potency of anesthetic agents’
Thank you for your comment. The paragraph has been changed based on your suggestion
Change ‘The use of cannabinoids has increased in a recreational and therapeutic way (2), to the knowledge of the authors, there are no reports of the effect of chronic administration of cannabinoids on the requirements of inhalation anesthetics.’ To ‘While the use of cannabinoids has increased in a recreational and therapeutic way (2), to the knowledge of the authors, there are no reports of the effect of chronic administration of cannabinoids on the requirements of inhalation anesthetics.’
Thank you for your comment. The paragraph has been changed based on your suggestion
Change ‘this investigation, we observed that 21 days of administering the synthetic cannabinoid WIN 55,212-2 increased the isoflurane MAC in rats’ to ‘this investigation, we observed that after 21 days of administering the synthetic cannabinoid WIN 55,212-2 MAC of isoflurane in rats increased.’
Thank you for your comment. The paragraph has been changed based on your suggestion
Change ‘WIN 55,212-2 cause an increase in the central levels of norepinephrine, Page and collaborators (20) showed that rats treated with WIN 55,212-2 for 8 days presented an increase in noradrenergic activity, furthermore they demonstrated that repeated administration of WIN 55,212-2 stimulates CB1 cannabinoid receptors in cell bodies of the locus coeruleus and in nerve terminals containing norepinephrine, generating an increase in norepinephrine efflux.’ To ‘WIN 55,212-2 causes an increase in the central levels of norepinephrine: Page and collaborators (20) showed that rats treated with WIN 55,212-2 for 8 days had an increase in noradrenergic activity. Furthermore they demonstrated that repeated administration of WIN 55,212-2 stimulates CB1 cannabinoid receptors in cell bodies of the locus coeruleus and in nerve terminals containing norepinephrine, generating an increase in norepinephrine efflux.’
Thank you for your comment. The paragraph has been changed based on your suggestion
Change ‘Norepinephrine levels in nervous terminals modulates MAC response of isoflurane as Miller et al’ to ‘Norepinephrine levels in nervous terminals modulate the MAC response of isoflurane as Miller et al’
Thank you for your comment. The paragraph has been changed based on your suggestion
Change ‘demonstrated in a previous work, in experiments following the administration of alpha-methyldopa, reserpine and iproniazid, they demonstrated that the requirements of inhalational anesthetics are related to norepinephrine levels.’ To ‘demonstrated in a previous work, in experiments following the administration of alpha-methyldopa, reserpine and iproniazid. These authors demonstrated that the requirements of inhalational anesthetics are related to norepinephrine levels.’
Thank you for your comment. The paragraph has been changed based on your suggestion
Change ‘It´s also known, diverse neurotransmitters such as norepinephrine, might manifest some of their actions by strongly inhibiting TWIK-related acid-sensitive K+ channels (TASK) and thus influence neuronal excitability.’ To ‘In addition, diverse neurotransmitters such as norepinephrine, may manifest some of their actions by strongly inhibiting TWIK-related acid-sensitive K+ channels (TASK) and thus influence neuronal excitability.’
Thank you for your comment. The paragraph has been changed based on your suggestion
Change ‘Similarly, Inhalation anesthetics activate and cannabinoid agonists inhibit TASK channels’ to ‘Similarly, inhalation anesthetics activate and cannabinoid agonists inhibit, TASK channels’
Thank you for your comment. The paragraph has been changed based on your suggestion
Change halothane MAC and cyclopropane MAC to MAC of halothane and MAC of cyclopropane
Thank you for your comment. The paragraph has been changed based on your suggestion
Change ‘for a lees period’ to ‘for a shorter period of time.’
Thank you for your comment. The paragraph has been changed based on your suggestion
Change ‘observed in de isoflurane MAC’ to ‘observed in MAC of isoflurane’
Thank you for your comment. The paragraph has been changed based on your suggestion

Reviewer 2 Report
- Page 2, section 2.1. Intubation using an “otoscope” appears incorrect. It was more likely a laryngoscope.
- Page 4, last 2 paragraphs. When referencing the differences with other studies, the decrease in MAC with acute administration is not surprising, as acute administration of other CNS depressants are expected to decrease MAC while chronic administration will increase MAC. This should be discussed.
- Page 5, last paragraph. The reason for the lack of change 8 days after stopping was not satisfactorily explained. One could expect some return to baseline. The authors should try to explain why this did not happen.
Author Response
Page 2, section 2.1. Intubation using an “otoscope” appears incorrect. It was more likely a laryngoscope.
Thank you for your comment.
We have placed the word laryngoscope
Page 4, last 2 paragraphs. When referencing the differences with other studies, the decrease in MAC with acute administration is not surprising, as acute administration of other CNS depressants are expected to decrease MAC while chronic administration will increase MAC. This should be discussed.
Thank you for your comment. We believe that this paragraph can explain your suggestion. Otherwise please tell us
Another explanation for the differences observed in MAC isoflurane can be through the observations made by Mechoulam et al. (29) and by Marciano et al. (30) which indicate that endocannabinoids have paradoxical effects on the central nervous system of mammals, since in some cases they generate an increase in neuronal excitability and in others they decrease it, depending on the dose administered. Similarly, they reported that cannabinoids cause short-term inhibitory effects on the release of glutamate; in contrast, prolonged stimulation of CB1 receptors by exogenous administration of cannabinoids could block the release of the inhibitory neurotransmitter, GABA (30).
Page 5, last paragraph. The reason for the lack of change 8 days after stopping was not satisfactorily explained. One could expect some return to baseline. The authors should try to explain why this did not happen.
Thank you for your comment. We believe that this paragraph can explain your suggestion.
Therefore, the increase in the isoflurane MAC caused by the administration of WIN 55,212-2 does not decrease after 8 days of stopping the administration of the cannabinoid, possibly caused by increase in noradrenergic activity sustained. This difference with the studies carried out by Page and collaborators (21), where a decrease in activity was observed at 8 days, could be a consequence of the longer exposure to the cannabinoid used in our study and possibly it will take more time for the decrease in noradrenergic activity since we administer it for 21 days. (31).

Reviewer 3 Report
GENERAL COMMENTS
Thank you for the opportunity to revise this interesting work on the effects of cannabinoids on MAC of isoflurane in rats. The topic is relevant and
As a general comment, this work would benefit from extensive language revision from a native English speaker. This would help to improve fluency, clarity, and readability. Please see below some specific comments.
ABSTRACT: the Background sentence should be reworded as this is part of the methods.
At the end of the introduction, it would be useful to state an hypothesis alongside the objectives/aims.
Paragraph 2.3: I recommend a clearer definition of the treatment groups, maybe using bullet points. A native English speaker may be able to help rewording this part in order to improve clarity and fluency. The description of the formulation should not be provided within the list of the treatment groups (either before or at the end of it).
Paragraph 2.4: if you expected a normal data distribution (data are reported as means and SD and parametric tests were used) the Shapiro Wilk test may not be the best test to evaluate data distribution.
RESULTS
The results paragraph should be expanded and needs further development. Please provide more data (e.g.: cardiovascular variables, duration of the anaesthetics, complications if any).
DISCUSSION
The first sentence should be in the introduction.
Author Response
Reviewer 3
GENERAL COMMENTS
Thank you for the opportunity to revise this interesting work on the effects of cannabinoids on MAC of isoflurane in rats.
The topic is relevant and
As a general comment, this work would benefit from extensive language revision from a native English speaker. This would help to improve fluency, clarity, and readability. Please see below some specific comments.
Thank you for your comment. We will address what is related to the English language based on the suggestions made by the editor
ABSTRACT: the Background sentence should be reworded as this is part of the methods.
Thank you for your comment, the Background sentence was changed based on another reviewer's suggestion, hoping that his suggestion can be answered
At the end of the introduction, it would be useful to state a hypothesis alongside the objectives/aims.
Thank you for your comment. Hypothesis is added based on your suggestion
The authors hypothesize that cannabinoids administered chronically have a different effect than that previously reported in acute administration of MAC.
Paragraph 2.3: I recommend a clearer definition of the treatment groups, maybe using bullet points. A native English speaker may be able to help rewording this part in order to improve clarity and fluency. The description of the formulation should not be provided within the list of the treatment groups (either before or at the end of it).
Thank you for your comment. We have placed the bullets based on your suggestion for better clarity.
The authors have doubts if you need to remove the term WIN-55,212-2 and place the word cannabinoid?
Paragraph 2.4: if you expected a normal data distribution (data are reported as means and SD and parametric tests were used) the Shapiro Wilk test may not be the best test to evaluate data distribution.
Thanks for your comment, we have chosen to use two Shapiro Wilk tests or Tuckey tests, and the Shapiro Wilk test provided greater statistical power.
RESULTS
The results paragraph should be expanded and needs further development. Please provide more data (e.g.: cardiovascular variables, duration of the anaesthetics, complications if any).
Thanks for your comment, the table with the physiological variables was added based on your suggestion.
DISCUSSION
The first sentence should be in the introduction.
Thanks for your comment, the definition of MAC has been moved to the introduction section based on your suggestion.

Round 2
Reviewer 3 Report
Thank you for answering my questions and addressing my concerns.
Please see below some minor comments:
I wonder if the minimum alveolar concentration would benefit from a brief explanation in the lay summary as I do not expect all the readers to be familiar with this term.
Line 76: oxygen flow RATE (please add “rate”)
Author Response
Please see below some minor comments:
I wonder if the minimum alveolar concentration would benefit from a brief explanation in the lay summary as I do not expect all the readers to be familiar with this term.
Thank you for your comment.
We have added the definition of MAC in the abstract based on your suggestion for better reader understanding.
Line 76: oxygen flow RATE (please add “rate”)
Thank you for your comment.
The word has been added